# The Transient Receptor Potential Vanilloid Type-2 (TRPV2) Ion Channels in Neurogenesis and Gliomagenesis: Cross-Talk between Transcription Factors and Signaling Molecules

**DOI:** 10.3390/cancers11030322

**Published:** 2019-03-06

**Authors:** Giorgio Santoni, Consuelo Amantini

**Affiliations:** 1School of Pharmacy, University of Camerino, Camerino 62032, Italy; 2School of Biosciences and Veterinary Medicine, University of Camerino, Camerino 62032, Italy; consuelo.amantini@unicam.it

**Keywords:** TRPV2, gliomagenesis, glioblastoma stem cells, neurogenesis, cancer stem cells, ion channels

## Abstract

Recently, the finding of cancer stem cells in brain tumors has increased the possibilities for advancing new therapeutic approaches with the aim to overcome the limits of current available treatments. In addition, a role for ion channels, particularly of TRP channels, in developing neurons as well as in brain cancer development and progression have been demonstrated. Herein, we focus on the latest advancements in understanding the role of TRPV2, a Ca^2+^ permeable channel belonging to the TRPV subfamily in neurogenesis and gliomagenesis. TRPV2 has been found to be expressed in both neural progenitor cells and glioblastoma stem/progenitor-like cells (GSCs). In developing neurons, post-translational modifications of TRPV2 (e.g., phosphorylation by ERK2) are required to stimulate Ca^2+^ signaling and nerve growth factor-mediated neurite outgrowth. TRPV2 overexpression also promotes GSC differentiation and reduces gliomagenesis in vitro and in vivo. In glioblastoma, TRPV2 inhibits survival and proliferation, and induces Fas/CD95-dependent apoptosis. Furthermore, by proteomic analysis, the identification of a TRPV2 interactome-based signature and its relation to glioblastoma progression/recurrence, high or low overall survival and drug resistance strongly suggest an important role of the TRPV2 channel as a potential biomarker in glioblastoma prognosis and therapy.

## 1. Expression of TRPV2 in Mammalian Central and Peripheral Nervous Systems

The Transient Receptor Potential Vanilloid (TRPV) subfamily (vanilloid receptors) includes ion channels involved in nociception and thermosensation (from TRPV1 to TRPV4), and in renal Ca^2+^ absorption/reabsorption (as TRPV5/6) [1].

Among them, TRPV2 is a non-selective cation channel showing Ca^2+^ permeability. The pharmacology of this channel is still insufficiently understood, and its structure has been resolved only recently by cryo-electron microscopy analysis (Figure 1) [2,3]. The channel structure shows six transmembrane domains, a putative pore-loop region, a cytoplasmic amino-terminus with three ankyrin-repeat domains, and a cytoplasmic carboxy terminus [4]. The transmembrane segment 6 (S6) is shown to be involved in gate opening with a rotation of the ankyrin-repeat domain coupled with pore opening via the TRP domain [3].

At the beginning, the TRPV2 receptor was considered a noxious temperature channel in sensory neurons (>52 °C), but then additional studies, performed in TRPV2 knockout mice, demonstrated that it displays other activities in vivo [5], functioning in normal sensory transduction, including temperature sensation. Moreover, it is well known that TRPV2 acts as a mechanosensor, osmosensor, lipid, and cannabinoid sensor. This receptor is ubiquitously expressed in particular in intracellular membranes, such as endosomes [6,7]. However, interestingly, TRPV2 expression was also identified in the plasma membrane of astrocytes [8], where it is activated by very high temperatures (>50 °C) and lysophosphatidylcholine, an endogenous TRPV2 lipid ligand. This suggests that in these cells, it may regulate neuronal activities in response to lipid metabolism [8].

TRPV2 is expressed in mammalian cells, both in the central and peripheral nervous system (CNS, PNS, respectively). In rats in particular, TRPV2 expression was identified in several areas of the brain, such as the forebrain, supraoptic nucleus (SON), magnocellular division of the paraventricular nucleus (PVN), organum vasculosum of the lamina terminalis, median preoptic nucleus subfornical, arcuate nucleus, medial forebrain bundle, cingulate cortex, and globus pallidus. Marked expression was observed in the hindbrain especially in the nucleus of the solitary tract, the hypoglossal nucleus, nucleus ambiguous, ventrolateral medulla, and in the area prostrema [9]. TRPV2 expression was detected in adult mice in the CNS, limited to some regions of the hypothalamus and brainstem [10]. Moreover, developing mouse dorsal root ganglia (DRG) and spinal motor neurons in mouse embryos express this channel [11]. A specific TRPV2 function in axon outgrowth has been suggested because its expression is mainly concentrated in myelinated A- and C-fiber sensory neurons [12], especially in developing growth cones. Shibasaki and coworkers demonstrated that during axon outgrowth, TRPV2 is activated by membrane stretch and, as a consequence, the intracellular Ca^2+^ concentration increases [11]. The majority of neurons positive for TRPV2 expression are peptidergic and TRPV2 is found to colocalize with both substance P and calcitonin gene related peptide (CGRP) [12]. Moreover, its activation by cannabinoid derivatives leads to CGRP release [13].

In humans, a complete neuroanatomical characterization of the TRPV2 expression in the CNS has been not conducted yet; the only available data describe the expression of TRPV2, both at mRNA and protein levels, in the cerebral cortex, hypothalamus, hippocampus, caudate and cerebellum. Moreover, recent findings demonstrate that TRPV2 is also expressed in human neural progenitor cells (hNPCs) and in normal human astrocytes (NHA) [14,15]. Moreover, it has been demonstrated that TRP channels represent the main route of Ca^2+^ entry in hNPC proliferating cells and the reduced spontaneous activity in differentiating hNPCs is associated with decreasedTRPV2 expression [15].

## 2. Role of TRPV2 Ca^2+^ Channel in Developing Neurons and Its Regulation by Signaling Pathways

During the NGF signaling pathway, responsible for neuronal cell differentiation, calcium plays an important role as second messenger. In fact, the resulting increase of intracellular Ca^2+^ concentration is then followed by changes in gene transcription and cell excitability [16]. Much evidence indicates a role of TRPV2 in the development of peripheral neurons and neurothrophin signaling [17]. The demonstration of the cross-talk between TRPV2 and nerve growth factor (NGF)-1 strongly suggests the contribution of this channel in neural development, in agreement with previous findings showing the involvement of TRPV2 in the CNS physiology [11,17]. The association between NGF pathways, fundamental for the differentiation of developing peripheral neurons, and TRPV2 has been demonstrated by studies performed on PC12 cells treated with NGF. In fact, NGF treatment induces an increased expression of theTRPV2 protein levels and, furthermore, PC12 cells, transfected with TRPV2 mutant channels, show a marked reduction in NGF-induced neurite outgrowth. Moreover, the knockdown of TRPV2 decreases neurite size in NGF-treated PC12 cells, suggesting that the NGF-induced TRPV2 upregulation leads to increase in neurite extension by altering the intracellular Ca^2+^ levels [17]. In a similar way, inhibition or silencing of TRPV2 reduces axon growth in primary DRG neurons [11]. Therefore, it has been suggested that the embryonic abnormalities and perinatal lethality observed in TRPV2-null mice may depend on the aberrant peripheral neuron development [5].

It has also been demonstrated that mechanical stimuli are able to activate TRPV2 by promoting axonal outgrowth in both developing neurons and PC12 cells [18]. The blockage of TRPV2 in astrocytes improves astrocyte-mediated neuroprotection after oxygen–glucose deprivation and re-oxygenation by increasing the NGF synthesis and secretion [19].

The NGF pathway starts with binding to its specific receptor TrkA at the axon terminal of developing neurons. The binding triggers receptor modifications and activation of the PI3-K and MAPK signaling [20]. After the formation of the NGF/TrkA complex and its internalization, the TrkA receptor moves by endosomes in the neuronal cell body, thanks to microtubules continuing to send signals [20,21]. Although TRPV2 functions have already been associated with PI3-K activation [22,23], no direct connection has been found at present between PI3-K signaling and NGF-induced TRPV2 upregulation. In contrast, pharmacological or siRNA-mediated down-regulation of MAPK signaling is able to reduce TRPV2 expression levels in primary developing neurons and PC12 cells [20,21]. The endosomal platforms, moving along the neuronal extensions, allow the NGF/TrkA complex and local signaling molecules to meet, stimulating changes in gene expression when the endosomes reach the nucleus [24]. In embryonic DRG neurons, the TRPV2 channel co-localizes with Rab7, as well as TrkA and pERK1/2, confirming that the channel and the components of the neurotrophin transduction pathway are expressed in endosomes. In vitro, post-transcriptional modifications, mediated by ERK phosphorylation, have been observed in the TRPV2 protein, suggesting that in neuritis, the TRPV2 activation and the resulting intracellular Ca^2+^ influx can be directly modulated by ERK. Moreover, specific TRPV2 residues, subjected to post-translational modifications, have been recognized; in particular, four serine residues, localized in the N and C terminal regions of the channel structure, are considered to be possible ERK substrates [17,25]. Among them, reported findings clarified that Ser6 and Ser760 represent phosphorylation sites in the TRPV2 molecule [26]. In fact, their mutation impairs the ability of ERK2 to phosphorylate the TRPV2 channel, causing inhibition of NGF-mediated neurite outgrowth, TRPV2 protein expression, and 2-APB-induced Ca^2+^ influx. The co-expression of TRPV2 and components of the transduction pathway in endosomes offers an efficient strategy for regulating Ca^2+^ signals and avoiding changes in neuronal excitability [27].

## 3. Expression and Function of TRPV2 Ca^2+^ Channels in Glioblastoma Stem/Progenitor-Like Cells (GSCs)

Malignant brain tumors, including glioblastoma multiforme (GBM), are known for their high degree of cellular heterogeneity, invasiveness, aggressiveness, and lethality. The hypothesis that gliomas originate as a consequence of the neoplastic transformation of differentiated glial cells has been considered the only true hypothesis for a long time [28]. Today, recent findings claim that stem cells are involved in the origin of different types of cancer, e.g., GBM [29,30] and, in support of this new idea, GSCs have been isolated from both human brain tumors and several glioma cell lines [31]. GSCs, derived from the transformation of the normal hNPCs, play a pivotal role in the development of glioblastoma malignancy [32]. HNPCs and glial progenitor cells are expressed in multiple regions of the adult brain. HNPCs, which are multipotent and self-renewal, have been isolated from the subventricular zone, the dentate gyrus, the hippocampus, and the subcortical white matter. In humans, a population of astrocytes with the ability to function as hNPCs is present in the subventricular region. In other adult mammals, the presence of glial progenitor cells capable of producing astrocytes and oligodendrocytes has been observed, and previous findings demonstrated that gliomagenesis is often linked with a particular area of the brain, such as the subventricular zone [30]. With that in mind, it is important to take into account that hNPCs, to acquire tumorigenic features, undergo aberrant and abnormal proliferative and developmental pathways.

New evidence shows that thermo-, mechano- and osmotic-sensitive cation channels regulate glioma growth and differentiation. In this regard, several TRP family members function as sensors that promote intracellular Ca^2+^ influx to transduce physical signals as temperature, osmolarity, and stretch. Changes in the TRPV expression levels, associated with the acquisition of enhanced proliferation and unusual differentiation, are often described during the neoplastic transformation of cells. In fact, signals generated by ion channels, aside from playing crucial roles in excitability and impulse conduction, also regulate proliferation, migration, and apoptosis of glioma cells. Since GSCs are able to respond to signals that induce maturation/differentiation, the interest in devising therapies that promote the differentiation of cancer stem cells is increasing [33].

In gliomas, recent findings indicate that cannabinoids like cannabidiol (CBD), which is able to bind both cannabinoid and TRPV2 receptors, inhibit gliomagenesis by stimulating the differentiation of GSCs. In this regard, GSCs express TRPV2 and, more interestingly, during their differentiation, the reduction of nestin levels is associated with a progressive enhancement of GFAP and TRPV2 expression [14]. The blockage of TRPV2 by using the TRPV antagonist, Ruthenium Red or TRPV2 silencing in GSCs during differentiation, markedly reduces the expression of GFAP and class III beta-tubulin (βIII-tubulin) expression. On the other hand, GSC proliferation, stimulated by phorbol-12-myristate-13-acetate treatment, is associated with reduced TRPV2 expression levels and the partial inhibition of astroglial differentiation. Moreover, TRPV2 overexpression promotes the upregulation of GFAP and βIII-tubulin levels, which is associated with the inhibition of proliferation and colony formation (Figure 2). The results, obtained from in vitro experiments using GSC cell lines, were also confirmed by in vivo tests. In fact, in a xenograft mouse model, the injection of TRPV2-overexpressing GSCs lead to the formation of smaller tumor mass, characterized by cancer cells, with a reduced mitotic index and a more mature glial phenotype with respect to the control counterpart (Figure 3) [14].

Thus, GCSs are now considered to be responsible for the malignant phenotype of GBM. For this reason, new therapeutic strategies promoting cell differentiation are required to eliminate the tumor-driving cell population involved in gliomagenesis and in the acquisition of chemoresistance [34,35]. CBD is in the list of new promising anti-cancer compounds, since it has been shown to inhibit GBM growth by stimulating glial differentiation and decreasing the GSCs’ efficiency in glioma formation [36,37].

In fact, CBD, via TRPV2 activation, triggers the GSC differentiation by activating an autophagic process and inhibiting the GSCs clonogenic capability. It is able to reduce in a TRPV2-dependent manner cell proliferation and survival [38], promoting cell death and enhancement of chemosensitivity in human GBM and other cancer types [36,37]. It was demonstrated in GSCs that CBD-induced TRPV2 activation leads to the activation of autophagy by stimulating the expression of several genes involved in the autophagic process and in the unfolded protein response. The autophagic pathway, stimulated by CBD/TRPV2, reduces cell viability, inhibits the proliferation rate, and causes cell cycle arrest in the G0/G1 phase. All these changes have also been associated with a marked increase in GFAP and βIII-tubulin expression and a reduction in stem cell marker levels such as CD133, Oct-4, SSEA-1, and nestin, leading to GSC differentiation [14]. In addition, AKT inhibition, or PTEN upregulation, is found in CBD-treated GSCs. The co-treatment with autophagy blockers inhibits these effects, suggesting that the autophagy is essential for the CBD-induced GSC differentiation. These data are also supported by findings demonstrating that the usage of the autophagy activator rapamycin promotes GSC differentiation, whereas 3-MA and BAF1, autophagic inhibitors, repress the serum-induced GSC differentiation [39].

It is well known that GSCs are resistant to conventional anti-cancer drugs such as Carmustine (BCNU) [40]. The combination of CBD with BCNU, by inducing apoptotic cell death, has proven useful in making GSCs much more sensitive to the action of BCNU. The enhancement of the GSC differentiation status increases the BCNU and Temozolomide (TMZ) chemosensitivity [41], and in glioma xenografts the growth of tumor is strongly reduced when TMZ is administered in combination with THC or with THC plus CBD [35]. In addition, the treatment of GSCs with CBD reduces the transcription levels of genes involved in chemoresistance, such as BCL-XL and CTDS mRNAs, and upregulates those responsible for the reestablishing of the apoptotic pathway as BAD and BAX [42].

## 4. The Transcription Factor Aml1/Runx1 Regulates the Proliferation and Differentiation of GSCs

Cancer stem cells have been identified in several cancers and, moreover, it is now known that functional ion channel currents are present in different types of stem cells. However, data concerning the expression and role of ion channels and their regulation at transcriptional and not-transcriptional levels in cancer stem cells are very limited [43]. Several polymodal ion channels are expressed in nociceptive sensory neurons [44] and their functions have been found to be controlled at transcriptional level. Among transcriptional factors regulating neuronal differentiation, Runx1, a Runt domain transcription factor also known in mammals as AML1 (acute myeloid leukemia 1), has been demonstrated to play a major role. In fact, it is expressed in embryonic neural tissues and participates in mammalian neurogenesis, particularly in the development of motor and sensory neurons (Figure 4) [45]. It has been shown to be widely expressed in nociceptors during embryonic development, but becomes limited to Ret-expressing nociceptors in adult animals [46]. Runx1 is involved in the regulation of the proliferation and differentiation of specific populations of neural progenitor cells. In particular, Runx1 coordinates the proliferation and differentiation of olfactory receptor neuron (ORN) precursor cells containing distinct stages of proliferating neural progenitor cells, like stem cells. In addition, the role of Runx1 in the transition from proliferating precursor to nestin^+^ neurons has been suggested. Moreover, exogenous Runx1 expression in cortical neural progenitor cells increases proliferation and neuronal differentiation [47]. In neurons, Runx1 controls the expression of ion channels and receptors (e.g., TRPV and TRPM channels, Na^+^-, H^+^-, ATP-gated channels, opioid receptors, and some G protein-coupled receptors) [46], and the expression of thermal receptors of the TRP channel family, including TRPV2, is abrogated or markedly reduced in adult Runx1^-/-^ mice (DRG). Consequently, Runx1 mutants show a delayed response to noxious heat stimuli and, because TRPV1 levels are significantly reduced, the pain responses stimulated by capsaicin are strongly inhibited [46].

Finally, in the cranial ganglia and hindbrain, Runx1 has been found to promote neuronal survival; by contrast, inactivation of Runx1 enhances neuronal cell death [48].

Mutations of Runx1, such as translocation, point mutation, or amplification, are often present in myeloid and lymphoid leukemia. Changes in its expression have been observed in cancers: For example, it is over-expressed in endometrial carcinoma [49] and down-regulated in gastric cancer [50]. In addition, it has been found that Runx1 expression correlates with tumor aggressiveness in astrocytomas [51] and gliomas [52]. Overexpression of Runx1 in P185 wild-type cells inhibits the BCR-ABL-induced proliferation and migration in vitro and leukemia genesis in vivo [53]. Moreover, the overexpression of RunX1 in NIH3T3 cells leads to neoplastic transformation [54].

In humans, a characteristic feature of the Runx1 gene is the generation of alternatively spliced transcripts. The human Runx1/AML1 gene generates three alternatively spliced variants, AML1a, AML1b, and AML1c. AML1b and AML1c are considered similar because they possess DNA-binding and transcriptional regulatory domains. Unlike AML1b and AML1c, AML1a is devoid of the transcriptional region, but maintains the DNA-binding domain. Evidence regarding isoform-specific functions comes from cell culture studies. AML1/Runx1 is thought to be involved in the balance between cell proliferation and differentiation [55]. Overexpression of AML1a (Runx1/p26 isoform) in murine myeloid cell lines abrogates the differentiation and stimulates the proliferation of cells. On the other hand, AML1b (Runx1/p49) mediates the opposite effect [56]. AML1a negatively regulates AML1b by binding CBFβ involved in the Runx1/DNA binding or competing for the DNA-binding sites [56]. Thus, the relative balance between the different AML1 isoforms drives the cellular fate (proliferation vs. differentiation) [57].

Recently, the expression of the AML-1a, AML-1b, and AML-1c spliced variants in different GSC lines has been reported [58]. Although the AML1/Runx1 variants are all expressed in GSCs, only AML-1a is upregulated in differentiated GSCs (D-GSCs) and consequently, it accumulates at a nuclear level. CBD, a TRPV2 agonist, enhances both AML-1a and TRPV2 expression in D-GSCs. At present, the specific involvement of AML-1a, compared to the other isoforms, in the survival and differentiation of GSCs is still unknown [59]. AML-1 enhances the expression of different ion channels belonging to the TRP family, such as TRPV1, TRPA1, and TRPM8 [60]. The TRPV2 overexpression and the inhibition of proliferation during GSC differentiation [36] require the binding of AML-1a to the TRPV2 promoter, since its inhibition markedly diminishes the TRPV2 protein expression. The knockdown of AML-1a by RNA interference during GSC differentiation reduces the expression of differentiation markers (e.g., GFAP, βIII-tubulin, and TRPV2) and triggers stemness, pluripotency, and proliferation.

The transcriptional activity of Runx1 is regulated by MAPK-mediated phosphorylation. Post-transcriptional events (ERK-dependent phosphorylation of Ser-249 and -266), that disrupt the Runx1/mSin3A interaction, activate Runx1 [61]. In agreement with these findings, it has been demonstrated in TRPV2-transfected GSCs that phorbol ester (PMA) reverts the TRPV2-induced inhibition of AML1a and increases the AML1b mRNA levels in an ERK-dependent manner [58]. Further studies will be required to completely understand the functional role of AML1a isoforms and the TRPV2/ERK pathway in the control of the balance between GSC proliferation and differentiation.

Knockout of the AML-1a by silencing (siAML1a) stimulates GSC proliferation through the induction of the Fibroblast growth factor (FGF) and Hedgehog pathway gene expression, including FGFR-2,-3,-4, GLI1, and GLI2, responsible for the initiation of glioma cell proliferation [62,63]. Increased siAML1a-GSC proliferation may also depend on enhancement of BMP and TGFβ genes (ACVR2A, BMPR1A, LTBP4, SMAD-6, and SMAD-9) [64,65]. In addition, the enhancement of the IL-6/IL-6R gene pathway (involved in GSC self-renewal and pluripotency [66], zinc finger E-box binding homeobox-2 (ZEB2) signaling, regulating E-cadherin expression, differentiation, proliferation, and invasion [67]) was found in siAML-1a-GSCs.

The understanding of the molecular mechanisms regulating self-renewal, proliferation, and differentiation of GSCs and the contribution of the different AML-1/Runx1 isoforms will improve the development of novel therapeutic strategies.

## 5. TRPV2 Channels in Glioblastoma Progression: TRPV2 Interactome-Based Signature as a Negative Prognostic Factor

GBM is the most devastating primary brain neoplasm with a median survival of 12–15 months [68,69]. Recent findings have demonstrated with RT-PCR and biochemical analyses the expression of the TRPV2 receptor in normal human astrocyte and glioblastoma tissues, with progressive reduction in TRPV2 expression as grade increases [70]. TRPV2 gene silencing in the high-grade U87MG GBM cell line increases the proliferation and resistance to apoptotic cell death, by up-regulating cyclin E1, cyclin-dependent kinase 2, E2F1 transcription factor 1, V-raf-1 murine leukemia viral oncogene homolog 1, and the apoptotic survival factor, Bcl-XL, as well as down-regulating Fas/CD95 and procaspase-8 mRNA expression. In agreement with these findings, pharmacological inhibition of the ERK pathway by using PD98059, a specific MEK inhibitor, reverts the TRPV2-mediated effects and stimulates glioma cell survival and proliferation in the U87MG GBM cell line [70].

The aggressive behavior of GBM, responsible for the short survival of patients, depends on high invasiveness and a high proliferation rate, as well as drug chemoresistance. TMZ, BCNU, and doxorubicin (DOXO) are usually used in GBM treatment, although their efficiency is partial. The recent data collected in this review indicate that the overcoming of chemoresistance in GBM patients could be obtained by treatment with cannabinoids. In this regard, by increasing TRPV2 expression and activity, CBD, a TRPV2 agonist, is able to enhance drug-uptake. The pore region of TRPV2 has been demonstrated to be critical for ion channel permeation and the resulting increase of glioma chemosensitivity and cytotoxic effects [36].

Several findings demonstrated that gene-based signatures’ analysis might produce a lot of new information in the understanding of the malignancy of GBM to improve its diagnosis, prognosis, and therapy [71,72,73]. A recent proteomic study identified a signature (TRPV2 + 22 proteins) significantly associated with overall survival of GBM [74]. This TRPV2 interactome is involved in nervous system diseases like brain tumors [74]. In this study, a set of proteins, including trafficking, transporter, and catalytic proteins, were identified as potential robust interactors of TRPV2. Among them, six proteins, KNJ10, SDC3, PLP1, ABR, FGF1, and PEBP1, were also demonstrated to be strongly associated with brain neoplasms. In this regard, the expression of KCNJ10, along with that of two other channels, KCNN4 and KCNB1, was found to be associated with overall survival (OS) in GBM. In particular, this genes-based signature predicts the prognosis of primary GBM patients and improves the GBM classification identifying a chemosensitive mesenchymal IDH1-wild-type subtype [73]. In addition, syndecan-3 (SDC3), a cell surface heparan sulfate proteoglycan, involved in mobility, migration, and invasion, was found to be overexpressed in GBM specimens and cell lines [75]. The expression of PLP1, a predominant component of myelin, is correlated with the percentage of the oligodendroglioma component in gliomas [76,77] and down-regulation/loss of the ABR tumor suppressor gene was found in gliomas and medulloblastoma [78,79]. Down-regulation of PEBP1, also known as the Raf-1 kinase inhibitor protein (RKIP), is associated with glioma progression [80,81], and it has been demonstrated that RKIP, by inhibiting MMP-2 and MMP-9 expression, markedly reduces the ability of glioma cells to migrate and invade [82]. Finally, it was shown that fibroblast growth factor 1 (FGF1), by activating FGF receptors, regulates cell proliferation, neurosphere formation, and neurogenesis. In addition, the FGF1/FGFR signaling, by increasing the AurA kinase expression, promotes the maintenance of stem cell features of GBM stem cells [83].

GBM patients show altered DNA repair pathways leading to chemotherapy resistance. Temozolomide sensitivity has been related to overexpression of O^6^-methylguanine methyltransferase (MGMT) and/or lack of DNA repair. Using the TRPV2-interactome signature, it has been demonstrated that patients with low OS risk show overexpression of MGMT and TRPV2 and down-regulation of base excision repair (BER) genes.

The expression of the TRPV2 interactome is significantly associated with the reduced survival of GBM patients. The TRPV2 interactome-based signature could be useful to discriminate among high- and low-risk GBM based on overall survival and to evaluate GBM progression, recurrence, and resistance to TMZ treatment [74].

## 6. Conclusion and Perspectives

The introduction of target therapy and the knowledge of molecular mechanisms regulating tumor development and progression have revolutionized cancer prognosis. In this regard, studies on the structure, expression and physiological/pharmacological functions of ion channels, belonging to the Transient Receptor Potential family, demonstrate that these channels play pivotal roles in cancer. Since they modulate specific stages of cancer progression, such as proliferation, invasion, migration, and chemosensitivity, they represent new interesting targets on which to base an innovative therapeutic strategy. In fact, changes in the expression and functions of many other components of this family like TRPC1, TRPC6, TRPM2, TRPM3, TRPM7, and TRPM8, in addition to TRPV2, which is the object of this review, have been demonstrated in GBM [84,85,86].

Findings derived from the integration of physiological and structural biology will open new perspectives on TRPV2 molecular mechanisms, especially on specific TRPV2 exogenous and endogenous modulators. In addition, the analysis of the TRPV2 interactome platform has already provided important information, such as protein–protein interactions implicated in cancer biology. Further studies on the TRPV2 interactome may bring about greater physiopathological understanding of TRPV2 functions.

Several reports demonstrate that CBD performs different actions, including anti-cancer activities, in a TRPV2-dependent and TRPV1- or cannabinoid receptor-independent manner [13,58]. CBD has recently been used in conjunction with chemoradiation therapy [87], in human patients affected by high-grade glioma to improve chemoradiation responses, suggesting that it is a good candidate for managing glioma by triggering the TRPV2 pathway.

In addition, four new chemical compounds were recently identified as selective TRPV2 inhibitors [88]. They have been shown to ameliorate cardiac dysfunction and prevent disease progression in the animal model of cardiomyopathy, as well as muscular dystrophy, by inhibiting the abnormal Ca^2+^-entry.

However, the development of methods identifying other agonists and/or antagonists of TRPV2 in a high-throughput screening fashion, is more necessary today than ever.

The discovery of specific TRPV2 compounds will provide the possibility to approach new therapeutic strategies with potential implications for different types of cancers and for CNS disorders.

## Figures and Tables

**Figure 1 cancers-11-00322-f001:**
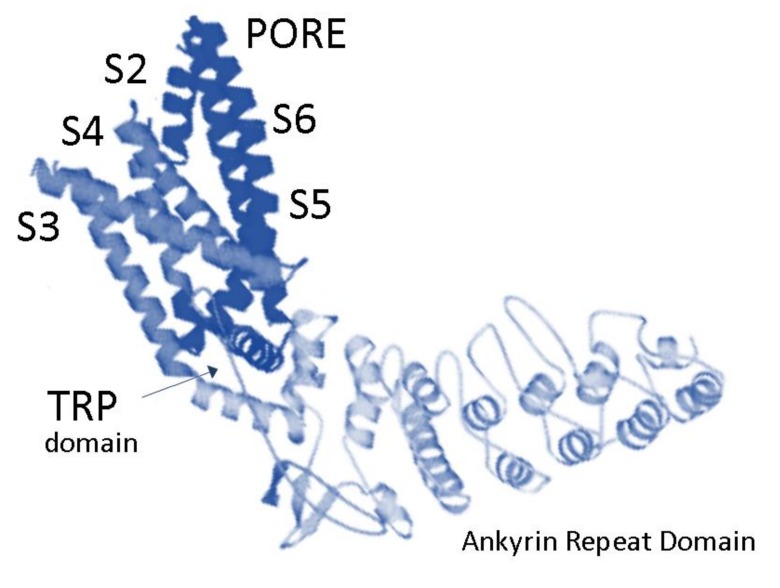
Overview of TRPV2 structural elements.

**Figure 2 cancers-11-00322-f002:**
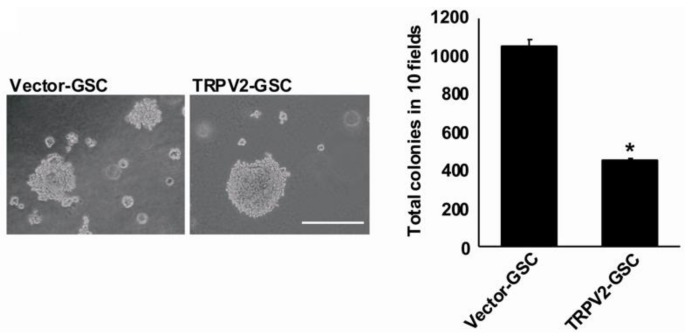
TRPV2 expression reduces glioblastoma stem/progenitor-like cell (GSC) proliferation. Representative phase-contrast photomicrographs show colony formation in vector- and TRPV2-transfected GSCs. The colony number was calculated, taking into account ten random fields. Data shown are representative of one of two separate experiments, * *p* < 0.01 vs. vector GSCs. Bar: 500 μm (figure is from [14]).

**Figure 3 cancers-11-00322-f003:**
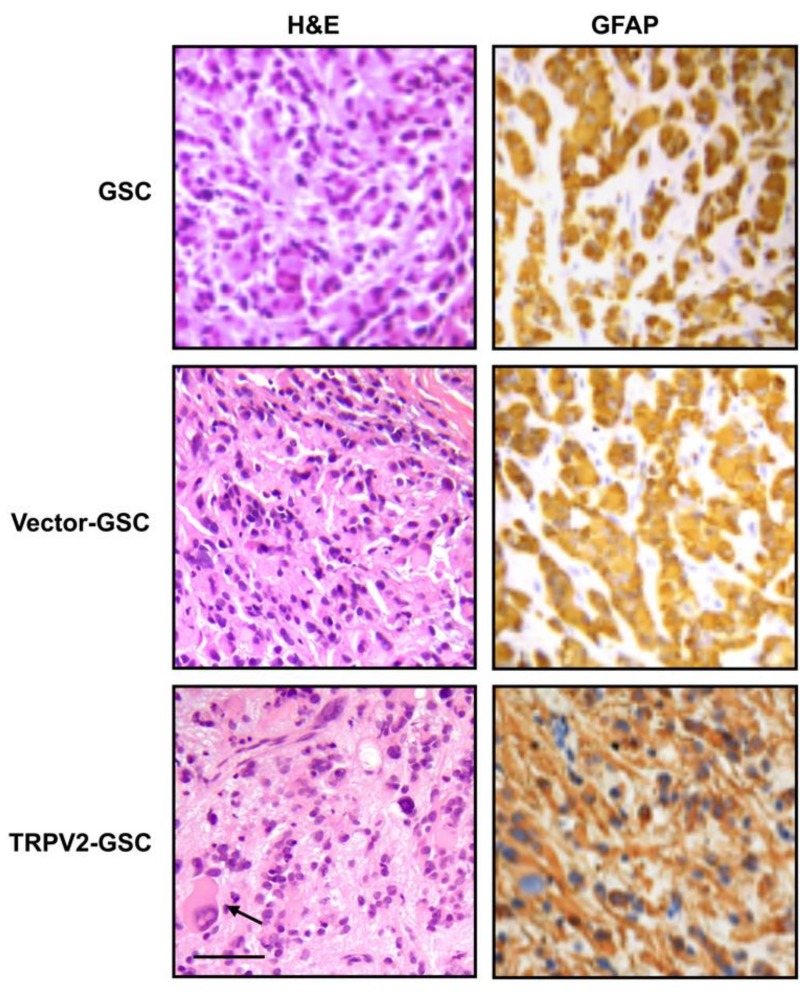
Enhancement of the astroglial phenotype is evident in tumors derived from transplanted TRPV2-transfected GSC lines. GFAP expression was analyzed in tumor xenograft sections stained with H & E. Bar: 50 μm. Arrow denotes multinucleated giant cells (figure is from [14]).

**Figure 4 cancers-11-00322-f004:**
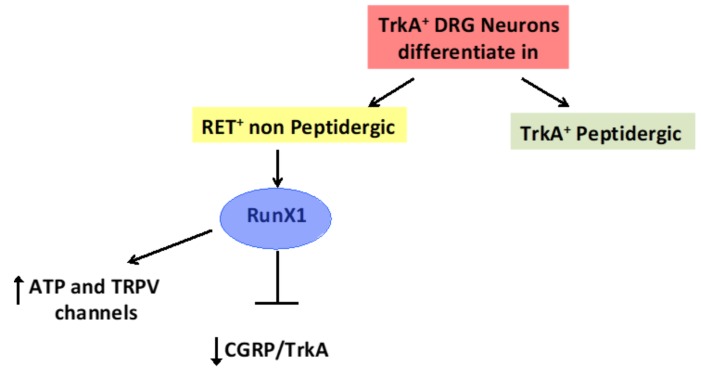
The differentiation of DRG neurons is Runx1-dependent. In Ret^+^ non-peptidergic neurons, Runx1 represses TrkA and the CGRP neuropeptide and promotes TRPV channel expression.

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
