# Peer review of "The Transient Receptor Potential Vanilloid Type-2 (TRPV2) Ion Channels in Neurogenesis and Gliomagenesis: Cross-Talk between Transcription Factors and Signaling Molecules"

_cancers, 2019, doi:10.3390/cancers11030322_

Round 1
Reviewer 1 Report
This is a detailed review on the expression and role of TRPV2 in neurogenesis/glioma-genesis. The initial sections were informative. However, the review is for a Cancer journal and I found the sections on TRPV2 and gliomagenesis/cancer a little underwhelming in the context of considering TRPV2 a viable candidate for chemotherapy – after all there is no selective pharmacology, correct? CBD clearly has a broad polypharmacology so will engage many effects beyond TRPV2. Also it was unclear how unique the role played by TRPV2 was versus other TRP channels as the authors take a very candidate-centric approach. It would have been of interest to expand upon the TRPV2 interactome candidates vis a vis a role in cell growth and cancer pathogenesis. Certainly, this is a useful contribution for specialists keen on studying TRPV2 in a neuronal pathological context.
Author Response
We agree with the reviewer that it is necessary to improve the knowledge about specific agonists/antagonist of TRPV2. However, several recent reports demonstrated that cannabidiol performs different actions, including anti-cancer activities, in a TRPV2-dependent and TRPV1 or cannabinoid receptor-independent manner; thus it represents a good candidate to manage glioma by triggering TRPV2 pathway. Moreover, new promising compounds, able to specifically target TRPV2, have been recently discovered.
To strengthen this point, the conclusion section has been improved and new references have been added.
Of course, other TRP channels have been found to be involved in glioma development and progression. However, we have focused the attention on TRPV2 in this review as the title shown. To avoid thinking that only TRPV2 has a role in glioma, we have added a sentence, including references, in the conclusion section by naming other members of the TRP family found involved in this tumor.
In addition, as suggested by the reviewer, we have improved the section related to TRPV2 interactome adding information about the most important genes of the interactome involved in glioma.
As suggested, the English language has been revised.
Reviewer 2 Report
The authors provide a substantial and good summary of TRPV2 channels and their involvement in neurogenesis and gliomagenesis.
I just have two minor comments to improve the quality of the manuscript:
(1) for better understanding for the reader, I would incorporate a figure showing the cryo-electron structure of TRPV2 - also including identified, important structures/residues/domains within TRPV2.
(2) there are some english style errors that should be corrected (for example: Ca2+ intracellular concentration should be changed to intracellular Ca2+ concentration; a role for ion channels to a role OF ion channels; several commas and so on)
Author Response
As suggested by the reviewer, a figure showing the cryo-electron structure of TRPV2 has been added. It is Figure 1.
In addition, as suggested, the English language has been revised to eliminate some English style errors.
This manuscript is a resubmission of an earlier submission. The following is a list of the peer review reports and author responses from that submission.